# Dinoflagellate Phosphopantetheinyl Transferase (PPTase) and Thiolation Domain Interactions Characterized Using a Modified Indigoidine Synthesizing Reporter

**DOI:** 10.3390/microorganisms10040687

**Published:** 2022-03-23

**Authors:** Ernest Williams, Tsvetan Bachvaroff, Allen Place

**Affiliations:** Institute of Marine and Environmental Technologies, University of Maryland Center for Environmental Science, 701 East Pratt St., Baltimore, MD 21202, USA; bachvarofft@umces.edu (T.B.); place@umces.edu (A.P.)

**Keywords:** dinoflagellate, PKS, phosphopantetheinyl transferase, toxin, BpsA, indigoidine, natural product

## Abstract

Photosynthetic dinoflagellates synthesize many toxic but also potential therapeutic compounds therapeutics via polyketide/non-ribosomal peptide synthesis, a common means of producing natural products in bacteria and fungi. Although canonical genes are identifiable in dinoflagellate transcriptomes, the biosynthetic pathways are obfuscated by high copy numbers and fractured synteny. This study focuses on the carrier domains that scaffold natural product synthesis (thiolation domains) and the phosphopantetheinyl transferases (PPTases) that thiolate these carriers. We replaced the thiolation domain of the indigoidine producing BpsA gene from *Streptomyces lavendulae* with those of three multidomain dinoflagellate transcripts and coexpressed these constructs with each of three dinoflagellate PPTases looking for specific pairings that would identify distinct pathways. Surprisingly, all three PPTases were able to activate all the thiolation domains from one transcript, although with differing levels of indigoidine produced, demonstrating an unusual lack of specificity. Unfortunately, constructs with the remaining thiolation domains produced almost no indigoidine and the thiolation domain for lipid synthesis could not be expressed in *E. coli*. These results combined with inconsistent protein expression for different PPTase/thiolation domain pairings present technical hurdles for future work. Despite these challenges, expression of catalytically active dinoflagellate proteins in *E. coli* is a novel and useful tool going forward.

## 1. Introduction

Dinoflagellates make a variety of natural products that have largely been identified based on their impact to human and animal health [1,2,3,4]. The actual biological and/or ecological roles are largely unknown and require further study. The exceptions include karlotoxin, the only toxin known to be actively released from the cell for prey capture and predator avoidance [5,6], and brevetoxin that likely functions as an indicator of redox state in the chloroplast [7,8]. This functional knowledge gap is exacerbated by a lack of a biosynthetic framework that would allow a more thorough cataloging of the natural products produced by dinoflagellates as well as insights into their evolution.

Natural product synthesis has been extensively studied in bacteria and fungi, yielding a mechanistic framework that operates as a series of modules with repeated chemistries followed by some modifications resulting in the final molecule. Essentially, small carboxylic acids are added to the thiol end of a phosphopantetheinate group attached to the serine of a carrier protein [9] via a condensation reaction that releases either carbon dioxide or water with prior activation by ATP [10,11,12]. These building blocks are then modified by subsequent reduction, methylation, carbon deletion, and other rarer reactions before the next carboxylic acid is added. In general, these are added by genetic modules comprised of single proteins with multiple functional domains or multiple cis-acting proteins brought together to form an enzymatic complex, although trans-acting elements are not uncommon [13,14,15] and substrates from multiple pathways can be combined [16,17].

Research into the biosynthesis of many natural products has relied heavily on the fact that gene arrangement is strongly predictive of a given natural products’ final structure. Unfortunately, dinoflagellate genomes are large and heavily duplicated [18], although mass spectrometry and NMR have been able to readily identify that dinoflagellate toxins have the hallmarks of classic natural product synthesis [19,20,21,22,23,24,25,26,27,28,29], with some exceptions [30]. Investigations into genes potentially involved in toxin synthesis have had some success [31,32,33], most notably in the separation of genes involved in natural product synthesis from the analogous synthesis of lipids [34,35,36,37] and the identification of multi-domain genes [38,39,40]. These multi-domain genes can then be used to further bin single domain genes into functional groups, although from here the waters become quite muddy with uneven gene copy numbers and the unprecedented duplication of genes related to lipid synthesis [41]. Thus, in many ways, sequence analysis has reached its limits in its ability to shed light on the synthesis of dinoflagellate natural products.

The aim of our project is to extend the current sequence-based knowledge into a biochemical based understanding of natural product synthesis by expressing dinoflagellate proteins in a heterologous system. An attractive target is the carrier protein called the thiolation domain that is activated by the attachment of the phosphopantetheinate group of coenzyme A by a phosphopantetheinyl transferase creating a free thiol moiety. This is the first rate limiting step in natural product synthesis and provides the substrate upon which the actual anabolism is performed by all of the catalytic enzymes. Generally, the activation of a thiolation domain by any phosphopantetheinyl transferase is highly specific and separates specific biosynthetic pathways, although the actual transfer of a phosphopantetheinyl group is not required for recognition of the transferase to a thiolation domain [42]. The thiolation domains of dinoflagellates can be readily separated into two main groups indicative of lipids and natural products [41]. Although the number of thiolation domains can total above one-hundred, the number of phosphopantetheinyl transferase activators is no more than three [43]. In addition, these activators have been expressed in *E. coli* [43] along with the indigoidine synthesis gene BpsA from *Streptomyces lavendulae* [44] that has been placed into an expression vector and characterized previously [45]. The rationale is that, if a given phosphopantetheinyl transferase can activate the thiolation domain of the BpsA reporter, then indigoidine will be produced. This pairing of activator and thiolation domain is a common method for determining specificity [46,47] and has been performed in some protists with a surprising promiscuity not found in bacteria and fungi [48,49]. This study advances previous work by replacing the thiolation domain of the BpsA reporter with several different dinoflagellate sequences to allow for the pairing of each activator with a multitude of potential phosphopantetheination sites. Although the integration of dinoflagellate sequence into the bacterially derived reporter was largely successful, there were several observations that led to the conclusion that this is qualitative only and that several artifacts of heterologous expression need to be overcome in future studies.

## 2. Materials and Methods

### 2.1. Reporter Modification

The BpsA reporter described in Owen et al. [45] was kindly obtained from the Ackerley lab at the University of Victoria in Wellington, New Zealand. The region encompassing the thiolation domain was amplified with the primers “BpsA_outF2” and “BpsA_outR2” listed in Table 1 at 500 nm final concentration and 10 µg of vector template using the Phusion high fidelity polymerase (New England Biolabs, Cambridge, MA, USA) as follows: Initial denaturation at 98 °C for 2′; followed by 40 cycles of denaturation at 95 °C for 15 s, annealing at 58 °C for 20″, and extension at 72 °C for 1′ 30″; and polishing at 72 °C for 5′. The amplified product, termed “BpsA_insert0”, was purified and sequenced at the BioAnalytical Services Laboratory (BASlab) at the Institute for Marine and Environmental Science in Baltimore MD on an Applied Biosystems 3130 XL. This sequence was used to design the remaining primers in Table 1 to insert a HindIII site in the 3′ end of the thiolation domain and an AflII site in the 5′ end as described in the primer name. The insertions result in the shift of arginine to a lysine at the HindIII site.

The HindIII and AflII sites were incorporated into the vector in a two-stage process. First, the HindIII site was created via two amplifications using “BpsA_outF2” with “BpsA_hindiiiR1” and “BpsA_outR2” with “BpsA_hindiiiF1” with the same reaction conditions as the thiolation domain amplification. The resultant products were purified using a DNA Clean and Concentrate-5 kit from Zymo research (Irvine, CA, USA) and eluted into 10 ul of distilled deionized water. Approximately 2.5 µg of product was digested with the HindIII-HF restriction enzyme from New England Biolabs for four hours at 37 °C, separated on an ethidium bromide impregnated 1% agarose gel in 0.5× TBE at 15 V/cm for 50 min, excised under ultraviolet illumination, and purified using a Monarch DNA Gel Extraction kit from New England Biolabs (Cambridge, MA, USA) as directed. The two digested fragments were then combined and ligated using a T4 ligase from Promega (Hercules, CA, USA) overnight at 18 °C. This product was then used as template for the second stage amplification using primers “BpsA_outF2” with “BpsA_afliiR1” and “BpsA_outR2” with “BspA_afliiF1” using the same conditions as the HindIII site amplification. This was purified, digested with AflII restriction enzyme from New England Biolabs, agarose gel purified, and combined and ligated in the same manner as the HindIII products resulting in “BpsA_insert1” (Figure 1).

Shown above is the BpsA gene from *Streptomyces lavendulae* originally published in Takahashi et al. [44] showing each of the domains with the thiolation domain marked with a “T”. The region surrounded by a blue box is expanded in the bottom showing the thiolation domain and the phosphopantetheinate transferase binding site as red boxes. The existing NotI and FspI restriction as well as the introduced AflII and HindIII sites are shown in red text. The primers used to isolate this region and attach the novel restriction sites are shown as green arrows above with the primer direction indicated by the arrow direction.

BpsA_insert1 was amplified using the same conditions as the original thiolation domain and purified using the DNA Clean and Concentrate-5 kit. This product as well as the original BpsA vector were double digested with the NotI-HF and FspI restriction enzymes from New England Biolabs at 37 °C overnight in cutsmart buffer followed by agarose gel purification and ligation as with the HindIII and AflII amplicons resulting in the BpsA2.1 vector. This was amplified using the Templiphi 100 kit from Cytiva (Marlborough, MA, USA) and cloned into *E. coli* JM109 from Promega (Hercules, CA, USA) according to the manufacturer’s directions. A selection of the resultant colonies was grown and the plasmid extracted for co-expression with each of the PPTases from *Amphidinium carterae* as in [43] to confirm activity.

### 2.2. Thiolation Domain Insertion and Co-Expression

The natural product associated thiolation domains [41] in three multi-domain transcripts (Figure 2) [37,38,41] from *A. carterae* were chosen for complementation in *E. coli* with the three *A. carterae* phosphopantetheinyl transferases (PPTases) that could activate them (Figure 3). These were termed “3KS” for the three ketosynthase domains present, “BurA” for its similarity in sequence and domain arrangement to the BurA gene in *Burkholderia species* [16], and “ZmaK” for the sequence similarity of the dinoflagellate adenylation domain in this transcript to the *Bacillus cereus* adenylation domain in the ZmaK cluster [17]. Each individual thiolation domain was named according to the transcript it was derived from the followed by a numeral indicating the order from 5′ to 3′ in the transcript, e.g., “3KS3” would be the third thiolation domain in the three ketosynthase domain containing transcript. The PPTase binding site amino acid sequence (Table 1) of each thiolation domain was codon optimized for expression in *E. coli* and ordered as an oligonucleotide from Integrated DNA Technologies (Coralville, IA, USA).

Individual modular synthase domains are shown at the top with example products for their reaction. In addition, Adenylation (A) and FSH1 serine hydrolases (FSH1) are shown for the multi-domain transcripts with examples of potential products included. The phosphopantetheinate group is shown as “P~P” with a single bind to a sulfur. “SL” refers to the dinoflagellate spliced leader sequence and is present if a spliced leader sequence has been verified.

Each oligonucleotide was synthesized with common linker sequences containing the AflII and HindIII restriction sites in the BpsA2.1 plasmid, one for the 5′ end, and one for the 3′ end (Table 1). Thus, each oligonucleotide consisted of the 5′ linker followed by the unique thiolation domain sequence and then the 3′ linker.

For each thiolation domain, the synthetic oligonucleotide as well as the BpsA2.1 plasmid were double digested with HindIII and AflII overnight at 37 °C in cutsmart buffer followed by agarose gel purification using a Monarch DNA Gel Extraction kit from New England Biolabs. The cut insert and plasmid were combined and ligated with a T4 ligase (Promega) at 18 °C overnight. Each ligated plasmid was amplified with the Templiphi 100 kit from Cytiva and cloned into *E. coli* JM109 from Promega according to the manufacturer’s directions. JM109 clones were sequenced to verify the presence of the dinoflagellate insert in the plasmid followed by alkaline extraction [51]. Plasmids were then cloned into chemically competent BL21(DE3) *E. coli* (Thermo Fisher, Waltham, MA, USA) along with one of the three PPTase activators (Figure 3) from *A. carterae* in a separate pet-20b plasmid [43] according to the directions for the competent cells and plated onto LB agar containing 100 µg/mL carbenicillin and 50 µg/mL spectinomycin (Sigma Aldrich, St. Louis, MO, USA) at 37 °C. Additionally, each PPTase vector and the thiolation domain vectors were individually cloned into BL21(DE3) to assess protein expression. The vectors for the PPTases were chosen to have a different replication sequence than the reporter to avoid conflicts during growth. Colonies were picked, grown in liquid media containing antibiotics overnight at 37 °C and stored at −80 °C with glycerol added to a final concentration of 12% *v*/*v*. For assessment of protein expression, glycerol stocks were used to inoculate 10 mL of LB in a 250 mL Erlenmeyer with appropriate antibiotics and grown overnight at 37 °C with shaking at 250 rpm. This was then diluted into 500 mL of LB media with antibiotics in a 2000 mL Erlenmeyer followed by a reduction of temperature to 30 °C and growth for 3 h with shaking. Protein expression was induced by the addition of 500 µL of 0.1 M IPTG followed by incubation at 25 °C for 3 h with shaking. Cells were spun at 10,000× *g* for 10 min at 4 °C, and the media was decanted. Cells were suspended in 20 mL of PBS at 4 °C with a bacterial protease inhibitor (Sigma Aldrich, St. Louis, MO, USA), and proteins were extracted in a French press at 20,000 local PSI followed by centrifugation at 10,000× *g* for 10 min at 4 °C to separate soluble and insoluble material. Insoluble proteins were recovered from the pellet by the addition of 6 M urea in equal volume to the supernatant. Heterologous proteins were purified with a 1 mL HiTrap Talon crude column (Cytiva, Marlborough, MA, USA) on an AKTA chromatography system with elution into 50 mM Tris with 250 mM imidazole. Proteins were separated by SDS-PAGE electrophoresis with 4–12% bis-tris gels (ThermoFisher, Waltham, MA, USA) and imaged with Imperial Coomassie stain (ThermoFisher, Waltham, MA, USA).

The *E. coli* clones containing one of the three *A. carterae* PPTases and one of the eight BpsA reporters with dinoflagellate thiolation domain sequence were each grown onto agar plates containing “autoinduction” media [52]. Colonies were grown at 25 °C for 48 h to allow for growth, protein expression, and indigoidine production. The plates were photographed, and each colony was assessed for dye production by measurement of grayscale density using image J (https://imagej.net/, accessed on 1 December 2018) with the space in between colonies as a baseline for background subtraction.

## 3. Results

### 3.1. Construct Generation and Domain Insertion

Following the generation of the BpsA2.1 vector with restriction sites flanking the phosphopantetheinyl transferase (PPTase) binding site of the thiolation domain, each of the eight dinoflagellate thiolation domain oligonucleotides were successfully inserted and verified by Sanger sequencing in both directions (not shown). Although each of the PPTases from *Amphidinium carterae* have previously been shown to interact with the wild type BpsA vector when co-expressed in *E. coli* [43], independent verification of protein production showed very different expression patterns for each of the three PPTases when expressed individually without the BpsA protein (Figure 4). In general, PPTase 3 showed high expression with protein in both the soluble fraction and the insoluble fraction recovered with 6 M urea following lysis of the *E. coli* host by French press. PPTase 2, however, was only visible in the insoluble fraction, and PPTase 1 had low expression in general.

An SDS-PAGE gel is shown for three *E. coli* clones containing the three *Amphidinium carterae* phosphopantetheinyl transferases following induction of protein expression with IPTG. Both the soluble (supernatant following French press isolation) and insoluble (Proteins retrieved from the pellet with 6 M urea) fractions are shown with arrows indicating the expected size of each protein based on the molecular weight marker designated as “Ladder” with kiloDaltons indicated.

Each of the constructs produced visible protein following his-tag purification (Figure 5)—in contrast to PPTase, the activators where PPTase 2 was not present in the soluble fraction in appreciable amounts. In order to explain how PPTase 2, which was previously shown to activate the wild type BpsA reporter [43], can function despite low apparent soluble production in *E. coli*, co-expression of PPTase 2 with both the BpsA2.1 vector without a heterologous insert as well as with the ZmaK1 insert was his-tag purified (Figure 6). The recoverable amount of the PPTase 2 activator as well as its substrate BpsA protein were higher in the original vector compared to the ZmaK1 insert containing vectors where both the reporter and the PPTase 2 activator were abundant in low amounts.

An SDS-PAGE gel is shown for the BpsA2.1 reporter with the standard sequence as well as one each of the triple-KS, ZmaK, and BurA inserts loaded with equivalent total protein. The size marker is shown on the left designated “Ladder” and an arrow shows the expected reporter size. The BurA1 protein was concentrated prior to imaging and shows several breakdown products.

### 3.2. Indigoidine Production

Following growth on autoinduction plates, co-expression of each of the PPTase activators with one of the BpsA2.1 vectors containing either the modified wild-type sequence or a dinoflagellate sequence insert resulted in similar growth for all colonies but indigoidine production in only some colonies (Figure 7). Background subtracted values show higher indigoidine production for the BpsA2.1 vector without inserts as well as with the triple KS inserts but not the ZmaK or BurA inserts with the exception of the combination of BurA2 and PPTase 3. In addition, the PPTase 2 activator pairings yielded consistently lower indigoidine production than PPTase 1 or 3 with the exception of the ZmaK2 insert that had low indigoidine production in all cases but was highest with PPTase 2. Other than the ZmaK2 insert, all BpsA pairings with the PPTase 3 activator resulted in higher indigoidine production relative to the PPTase 1 or 2 activators. Indigoidine production was also performed with each insert along with the PcpS gene, a bacterial PPTase from *Pseudomonas aeruginosa*, and a common control gene for phosphopantetheination, but indigoidine was only produced in appreciable amounts with the reporter without dinoflagellate inserts compared to low or almost no production with the dinoflagellate sequences (Appendix A).

The graph shows the relative darkness of each colony to a black pixel on the *y*-axis resulting from the production of indigoidine in *E. coli* upon the co-expression of the BpsA gene containing a dinoflagellate thiolation domain and a dinoflagellate phosphopantetheinyl transferase (PPTase) on separate vectors. The thiolation domain is indicated on the *x*-axis including wild type sequence (BpsA), as well as the *Amphidinium carterae* triple KS (KS), ZmaK-like (ZmaK), and BurA-like (BurA) transcript sequences with a numeral indicating the particular thiolation domain from the N to C terminus. The *z*-axis indicates which clade of PPTase sequence was co-expressed from Williams et al. [43]. The actual plates with induced expression are shown in the upper right in the same orientation as the graph *x* and *z*-axes for reference.

## 4. Discussion

Genetic tools such as knockdowns, knockouts, and knockins can be powerful means to determine gene function by looking for phenotype changes following a change in the expression of that gene. While these techniques are well established in many bacteria, fungi, and vertebrates, there are many branches of the tree of life where genetic techniques are not well developed for a variety of reasons. Protists, in particular, representing a huge amount of eukaryotic evolutionary diversity, have lagged behind, although new sequencing technologies have provided a much-needed boost in investigative power [53], resulting in several partial genomes for dinoflagellates in particular [54,55,56,57,58]. Some genetic modifications of dinoflagellates have been successful for the purpose of harnessing the unique lipids of marine protists [59] focusing on knockdowns and knockouts [60,61] but also to investigate the unique biology of dinoflagellates [62]. Success using these techniques is generally limited due to the high copy number of many dinoflagellate genes [18], especially for precision techniques such as CRISPR/Cas9. A gene knock in has been successful for a dinoflagellate chloroplast [63], but gene addition to the nucleus has remained elusive due to the post-transcriptional nature of dinoflagellate gene expression [64,65,66], making traditional promoter driven approaches unusable. Heterologous expression of dinoflagellate genes in more tractable systems is one way of getting around the biological difficulties of dinoflagellate genetics. This would allow for more direct biochemical analysis of dinoflagellate proteins and has been surprisingly successful with the documented expression of dinoflagellate proteins in mammalian cells [67].

In many ways, this study is a bridge between dinoflagellate biology, focusing on ecology and species diversity, and natural product research, a biochemical approach to discovering and harnessing useful compounds. This marriage seems a foregone conclusion given that dinoflagellates make compounds that affect human health negatively [4] but potentially positively as therapeutic agents, with neosaxitoxin being the first phycotoxin to be used clinically [68]. The difficult biology of dinoflagellates has made this area of research slow-moving although with the production of the polyunsaturated fatty acid DHA from *Crypthecodinium cohnii* [69] and the subsequent efforts for genetic engineering [60] being a notable exception. On the other hand, the natural product world has a great wealth of experience to draw from. Domain replacement has been a common technique to identify substrate specificity or produce novel compounds [70,71,72,73,74,75,76] and the BpsA gene itself has also been used extensively [77,78,79,80]. In the future, techniques such as NMR and MS based omics approaches can further inform biochemical validation.

Elucidating the function of genes involved in the synthesis of dinoflagellate toxins is especially tricky, since, like most natural products, the synthesis is inherently modular in nature [11], resulting in a large copy number of nearly identical genes. The availability of transcriptomes has helped greatly in cataloging and identifying the genes most likely to be involved in toxin synthesis [31,35,37,39]. While the catalytic genes for toxin synthesis are numerous and similar in sequence, the thiolation domains are less numerous and easily split into those likely to make lipids versus natural products by sequence alone [41]. When considering the phosphopantetheinyl transferases (PPTases) that activate these thiolation domains, which have fewer than three types and a low copy number [43], the number of combinations becomes tractable for biochemical analysis. Thus, this study aims to begin a bottom-up approach, where the specificity of a PPTase for a particular pathway, vis-à-vis that pathway’s thiolation domains, can be exploited to isolate toxin synthesis pathways once that specificity has been identified, although there appears to be some unusual promiscuity in protists [49].

Technically, this study represents a step forward as the first example of a catalytically active reporter containing dinoflagellate genes. Natural product synthesis can be quite complicated and dinoflagellate toxins are much larger than most natural products [30]. It is also unclear how many natural products dinoflagellates make since efforts have been so heavily focused towards toxins. In bacteria and fungi, the canon that each biosynthetic pathway for a natural product or lipid has a specific PPTase has been a useful tool in identifying and characterizing each pathway. In dinoflagellates, pathways are very difficult to identify because of the lack of gene synteny and a mathematical problem of copy number, e.g., the number of dehydratase domains is frequently lower than the number of enoyl reductases, even though the dehydratase reaction must occur before the reductase reaction [41]. Thus, biochemical methods like those presented here are necessary to proceed further in identifying natural product biosynthetic pathways in dinoflagellates. While not every combination of an insert containing reporter with a PPTase activator produced indigoidine, each insert was able to produce indigoidine with at least one of the PPTases (Figure 7). Some assumptions necessary for semi-quantitative measurements of indigoidine synthetic potential are that each transformed *E. coli* strain has the same plasmid copy number and expresses each protein equivalently. While the former is likely true, given that equivalent amounts of plasmid were used in the transformation, the latter is certainly false given the differential expression based on PPTase type and reporter pairing (Figure 5 and Figure 6). The *E. coli* colonies were allowed to induce expression and produce indigoidine for 48 h, and, not shown here due to a loss of quantitation, would eventually become dark in color for most insert/activator combinations. This immediately leads to the conclusion that all forms of the reporter are capable of producing indigoidine and that there is some possible phosphopantethienation by each dinoflagellate PPTase for any given thiolation domain, another example of promiscuous PPTase binding in protists [49]. This also verifies the prediction that these enzymes transfer the phosphopantetheinate group and not other moieties, as has been shown in rare cases [50]. In terms of efficiency, PPTase 2 generally produced the lowest amount of indigoidine for all inserts in 48 h, with the exception of ZmaK2, although the variability in the production of soluble protein (Figure 6) makes this result almost certainly spurious. However, if the poor performance of PPTase 2 when combined with these natural product associated thiolation domains is accurate, this leaves open the possibility that PPTase 2 may have more favorable interactions with the acyl carrier protein, the thiolation domain responsible for lipid synthesis. One of the biggest issues with this study is that the dinoflagellate acyl carrier protein is nearly identical to the *E. coli* sequence at the insert site (Figure 3). This construct was not able to be expressed in *E. coli* likely due to toxicity. Thus, comparisons between the thiolation domains presented in this study and the acyl carrier protein would be more appropriate for in-vitro based studies or co-expression studies in another host assuming that the issue of toxicity does not still exist. In contrast to the previous study of dinoflagellate PPTases [43], PPTase 1 generally had similar indigoidine production when compared to PPTase 3 instead of the much lower production previously shown. This is likely due to the much longer autoinduction based protein expression in this study compared to the short term IPTG based expression in the 2020 study.

There were some obvious differences observed between the triple-KS inserts and the ZmaK or BurA inserts in terms of the indigoidine produced (Figure 7). The triple-KS inserts had consistently high indigoidine production, and the triple-KS transcript can also be found in the more basal syndinian dinoflagellate *Hematodinium sp.* [81], a parasite of crustaceans. The BurA-like and ZmaK-like genes on the other hand are not found in any syndinian transcriptomes to date and are very similar in sequence to bacterial genes, making horizontal transfer a likely origin. The results presented here may indicate that, at least for the *Amphidinium carterae* PPTases, the ability to activate the BurA and ZmaK inserts is sub-optimal. This was suggested in the Williams et al. 2021 study on the thiolation domains of dinoflagellate domains based on the observation that many of the ZmaK sequences lie outside the cluster of natural product associated domains with the more basal sequence the furthest away, indicating that convergent evolution may be an active force. Thus, PPTases from more distal species of dinoflagellates may be better at activating the BurA and ZmaK insert containing reporters than the *A. carterae* based PPTases used here. It could also be that the BurA and ZmaK based sequences themselves are interfering with the reporter’s ability to produce indigoidine given that the thiolation domain acts as an intermediary for all other domains. The BurA and ZmaK sequences may be sterically interfering with the other domains of the BpsA protein, reducing its overall efficiency.

## 5. Conclusions

The modification and deployment of this indigoidine synthesizing reporter for dinoflagellate genes allows for new approaches to studying genes that may be involved in toxin synthesis. Specifically, methods that can validate suspected interactions during natural product synthesis are crucial in overcoming the limitations of sequence-based annotations. Here, domain substitution has been used to demonstrate the broad substrate recognition of dinoflagellate PPTases that can help explain gene losses and gains throughout dinoflagellate evolution that do not correlate with a loss or gain in toxin synthesis. There are also several not unexpected lessons to be learned from these results. First and foremost is that, despite the huge evolutionary distance between dinoflagellates and the *E. coli* heterologous host, some dinoflagellate genes simply cannot be expressed in this system, likely due to host interactions and resultant toxicity. This is both intriguing and frustrating that genes involved in lipid synthesis can be so conserved in such a dynamic group of organisms like dinoflagellates. Perhaps some comparisons to syndinian dinoflagellates that parasitize and absorb lipids from dinoflagellate hosts can shed light on how these genes function and have been modified over time. A second consideration is the dynamic nature of expression of the PPTases depending on the thiolation domain it is paired with. This renders these results qualitative at best and implies that in vitro methods, while more challenging, are likely more accurate comparisons of PPTase activity. Finally, the lack of indigoidine synthesis for the BpsA vectors containing the ZmaK and BurA inserts is suspicious and may be the result of steric hindrance rather than a lack of PPTase activation. A general conclusion from the activation of all the thiolation domain sequences used here by all three PPTases is that the substrate specificity observed in bacteria and fungi may be the exception in protists rather than the rule. The mechanism that dinoflagellates use to regulate the initiation of lipid and natural product synthesis is an important topic for future study. Targeted knockdowns are another avenue going forward to validate these results since broad substrate recognition shown here does not mean that specific natural product pathways are not physically separated in situ. In addition, knockdowns of acyl transferases and thioesterases that join and terminate major portions of natural product synthesis as proposed in Van Wagoner et al. 2014 [30] can help to isolate specific pathways.

## Figures and Tables

**Figure 1 microorganisms-10-00687-f001:**
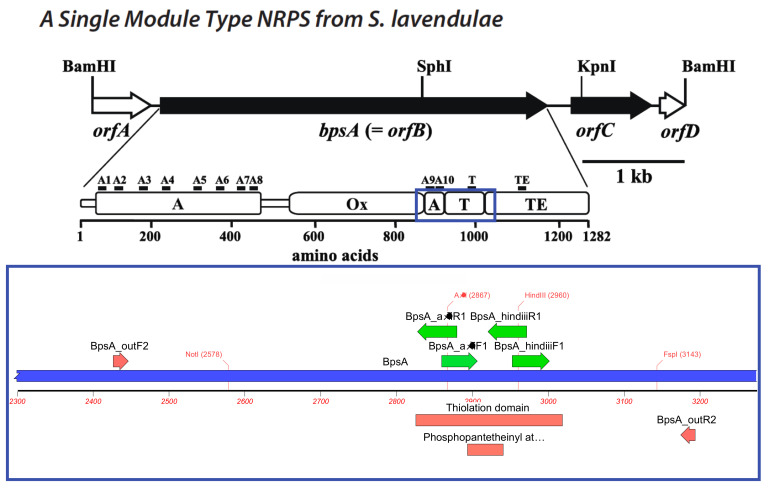
A modification of the BpsA to allow the insertion of dinoflagellate thiolation domain sequences.

**Figure 2 microorganisms-10-00687-f002:**
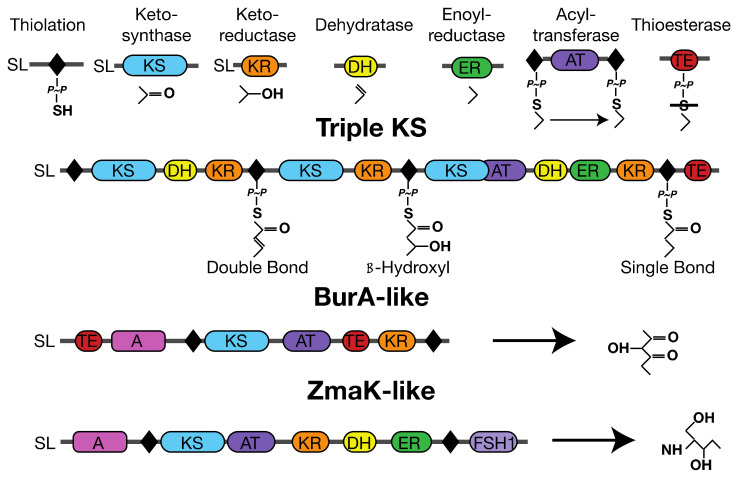
Domain arrangement of *A. carterae* transcripts containing thiolation domains used in this study.

**Figure 3 microorganisms-10-00687-f003:**
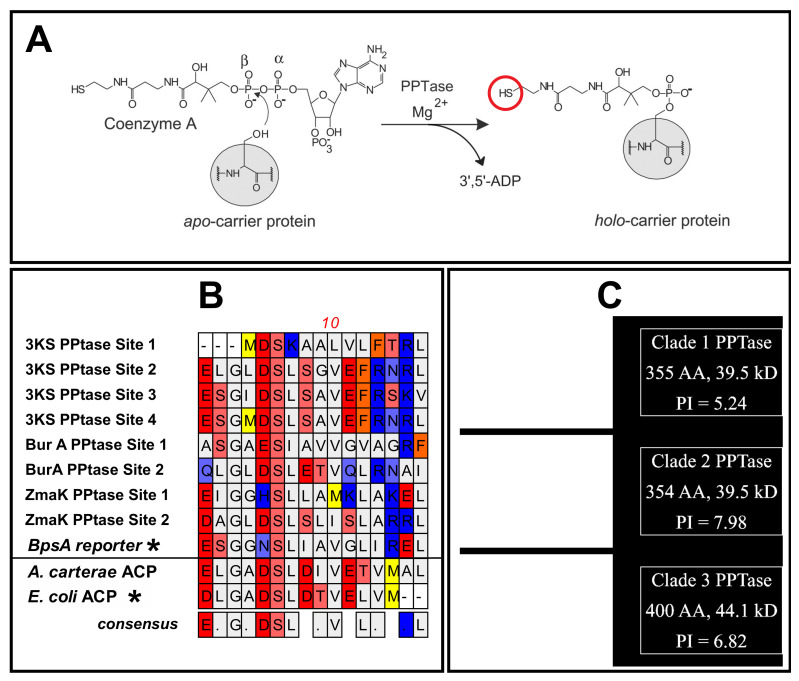
A mechanism of phosphopantetheination and the dinoflagellate thiolation domains used in this study; (**A**) a diagram of the phosphopantetheination reaction from Finking et al. 2002 [50] showing the phosphopantetheinate arm of coenzyme A attachment to the serine of a carrier protein or domain resulting in a free thiol group (red circle); (**B**) the amino acid sequences of the thiolation domains from *A. carterae* used in this study except those marked with a “*” that are from the *S. lavendulae* isolated BpsA gene and the acyl carrier protein (ACP) from *E. coli*. Sequences above the line are theorized to be involved in natural product synthesis while those below the line are for lipid synthesis; (**C**) the predicted folding for the three phosphopantetheinate transferases from *A. carterae*.

**Figure 4 microorganisms-10-00687-f004:**
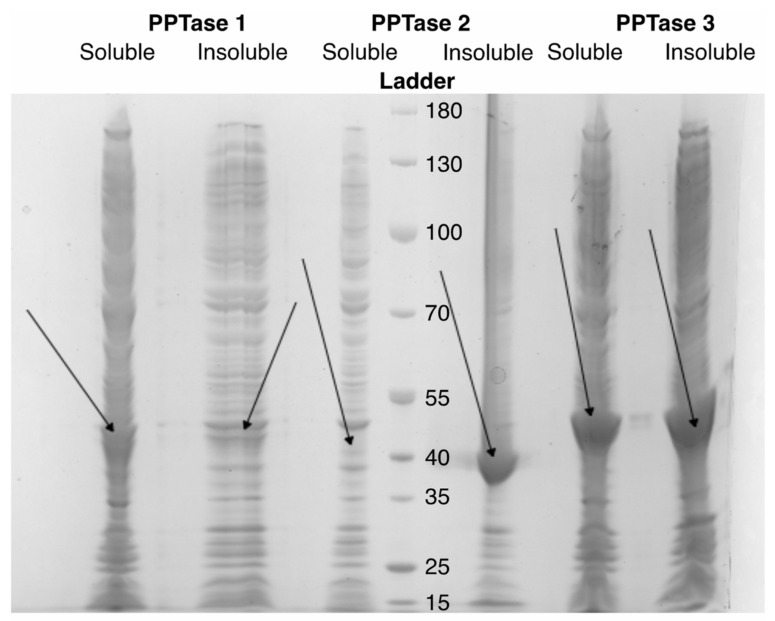
Soluble and insoluble lysates from *E. coli* following induction of phosphopantetheinyl transferase expression.

**Figure 5 microorganisms-10-00687-f005:**
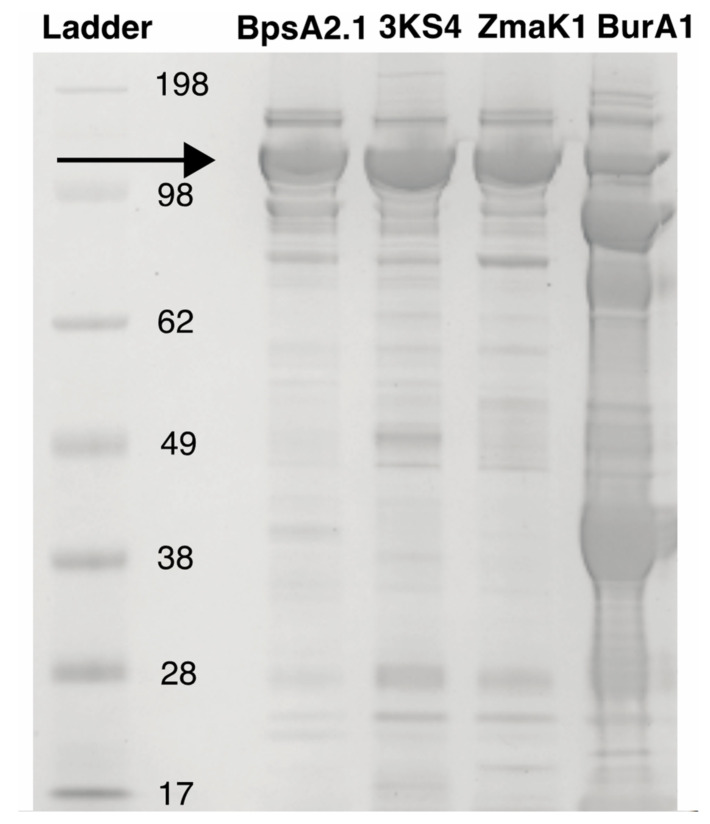
His-tag purified BpsA reporter.

**Figure 6 microorganisms-10-00687-f006:**
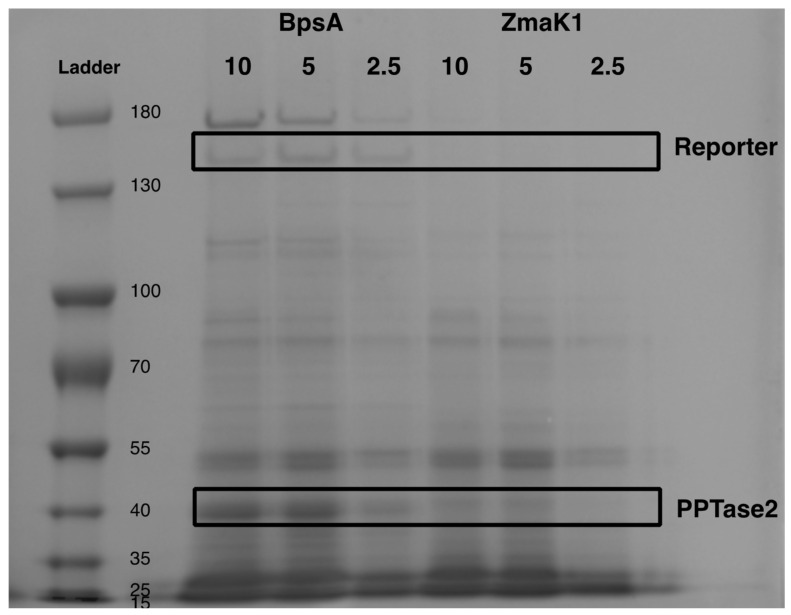
PPtase2 expression with BpsA reporter standard insert and ZmaK insert. An SDS-PAGE gel is shown for a co-expression of PPTase 2 with either the standard BpsA2.1 sequence or with the ZmaK1 insert following French press lysis and removable of insoluble material by centrifugation. The expected sizes for the reporter BpsA protein as well as the PPTase protein are highlighted with black boxes according to the expected size shown on the left with the size standard marked as “Ladder”. The load volumes are shown at the top of each well in microliters from equivalent *E. coli* cultures.

**Figure 7 microorganisms-10-00687-f007:**
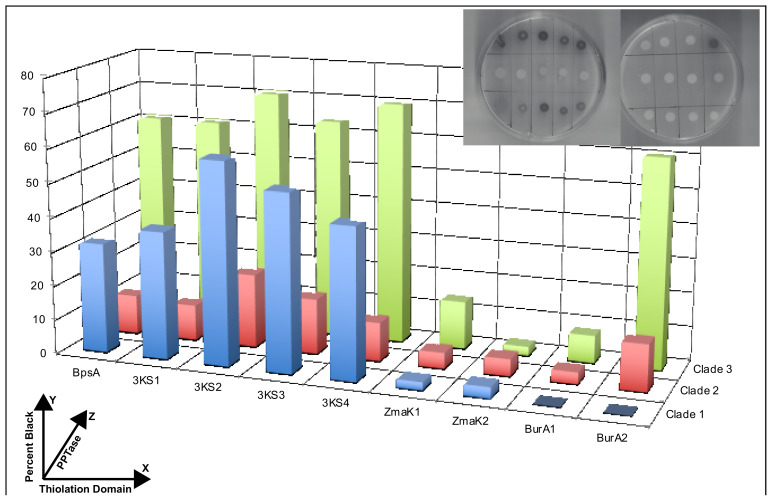
Indigoidine synthesis in *E. coli* from the co-expression of dinoflagellate phosphopantetheinyl transferases and the BpsA gene with a dinoflagellate thiolation domain.

**Table 1 microorganisms-10-00687-t001:** Synthetic primers and inserts used in reporter modification.

Primers			
Primer Name	Sequence 5′:3′	Length	Annealing °C
BpsA_outF2	TCCAGCACCTGATGATGAAC	20	58.4
BpsA_outR2	CTGGATGCCGTAGAACGAG	19	59.5
BpsAhindiiiR1	﻿GACGCCAAGCTTCGCGTTGAGCTCGCGGACGAGGCCGACGGCGATCAGCGA	51	91.1
BpsAhindiiiF1	﻿CAACGCGAAGCTTGGCGTCTCCCTGCCGCTGCAGAGCGTCCTGGAGTCC	49	89.6
BpsAafliiR1	CTCGCGCTTAAGGGCCTTCTCCCAGACCGCCGCGATCTCCTTCTCCGT	48	88.5
BpsAafliiF1	﻿AGAAGGCCCTTAAGCGCGAGAACGCCTCCGTCCAGGACGACTTCTTCG	48	86.4
**Inserts**			
Insert Name	Sequence 5′:3′	Binding Site Amino Acid	
3KS_1	GAATCGGGCATGGACTCAAAAGCAGCCCTTGTTCTG	**ESG**MDSKAALVL	
3KS_2	GAATTGGGCTTAGATTCTTTGTCCGGCGTTGAATTT	ELGLDSLSGVEF	
3KS_3	GAAAGCGGAATTGATTCCTTGTCTGCAGTAGAGTTT	ESGIDSLSAVEF	
3KS_4	GAGAGTGGCATGGACTCATTATCTGCCGTCGAGTTT	ESGMDSLSAVEF	
BurA_1	GCT TCA GGT GCA GAA TCT ATC GCT GTC GTG GGC GTG	ASGAESIAVVGV	
BurA_2	CAA TTA GGA TTA GAC AGC TTG GAA ACC GTT CAA CTG	QLGLDSLETVQL	
ZmaK_1	GAA ATC GGT GGG CAC TCG CTG TTA GCA ATG AAA CTT	EIGGHSLLAMKL	
ZmaK_2	GAT GCC GGG TTA GAT AGC TTA TCC TTA ATT AGC TTA	DAGLDSLSLISL	
5′ Linker ^†^	﻿AGAAGGCCCTTAAGCGCGAGAACGCCTCCGTCCAGGACGACTTCTTC		
3′ Linker ^†^	﻿GTCCGCGAGCTCAACGCGAAGCTTGGCGTC﻿TCCCTGCCGCTG		

“^†^” denotes common linkers to all other inserts and were placed at the 5′ and 3′ ends during synthesis as indicated. The “3KS_1” sequence shown in bold is the wild type sequence included to ensure consistent insert size.

## Data Availability

Not applicable.

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
