# Peer review of "Dinoflagellate Phosphopantetheinyl Transferase (PPTase) and Thiolation Domain Interactions Characterized Using a Modified Indigoidine Synthesizing Reporter"

_microorganisms, 2022, doi:10.3390/microorganisms10040687_

Round 1

Reviewer 1 Report

Microorganisms (Manuscript ID: microorganisms-1585421), Comments to the Authors: Title: Dinoflagellate Phosphopantetheinyl Transferase (PPTase) and Thiolation Domain Interactions Characterized Using a Modified Indigoidine Synthesizing Reporter Comments The submitted manuscript discussed the modification of the indigoidine producing bacterial BpsA reporter with dinoflagellate sequences in its thiolation domain. The amount of indigoidine produced following co-expression in E. coli with one of three Amphidinium carterae PPTases describes the specificity of each pairing. Although indigoidine production was evident there were some caveats, most notable that the acyl carrier protein for synthesizing lipids could not be expressed, likely because of host toxicity. The amount of soluble protein for one PPTase varied depending on the reporter it was expressed with making these results qualitative but not reliably quantitative. I think the submitted manuscript can be accepted after the authors respond to the following comments: 1. The abstract is disorganized and should be rephrased to provide clear picture of the manuscript results and show the value of the work. 2. The authors should clearly indicate the application of their findings for the synthesis of natural products. 3. The authors should add few statements on the future perspectives of their work. 4. The authors should compare their work with previous research to show the value of the submitted manuscript. 5. The authors should rephrase the conclusion section and remove refernces to the figures.

Author Response

Response to reviewer

1)The abstract has been rewritten with several aims

            a)To reduce the amount of background information

            b)To specifically state the aims of the study

            c)To more directly summarize specific results

            d)To identify the usefulness of this study

2)The abstract, discussion and conclusion have been modified to more specifically state the application of this work to natural product research in dinoflagellates and to the natural product community as a whole.

3)Specific examples of future work have been added to the conclusions section to give more perspective on how to move forward.

4)An additional section has been added to the discussion about the intersection of natural products and dinoflagellate toxin research to give a backdrop for this study. In particular, examples of domain replacement and the BpsA reporter itself are mentioned to give context to the methodology used here.  We apologize for this oversight and appreciate how we had overlooked the members of the natural product research community that may have been interested in this work and who provided the historic developments enabling this study.

5)The conclusion section has been reworded to remove specific references to the data and make more general statements. Also, references to figures have been removed.

Reviewer 2 Report

This study aimed to contribute to the elucidation of the genes function involved in the synthesis of dinoflagellate toxins. This is something especially tricky since the synthesis of natural products results in large copy number of nearly identical genes. The authors chose the thiolation domains of dinoflagellates, because i) thiolation is one initial step in the anabolism driving to toxin production, i) it can be readily separated into two main groups indicative of lipids and natural products, and iii) there are only 1 to 3 phosphopantetheinyl transferase activators.

This study advances previous work by replacing the thiolation domain of the BpsA reporter (from a bacteria) with eight different dinoflagellate sequences to allow for the pairing of each activator with a multitude of potential phosphopantetheination sites.

The study successfully managed to integrate dinoflagellate sequences into the bacterially derived reporter. However, this result was mainly qualitative and several artifacts of heterologous expression need to be overcome in future studies.

I specially appreciate the detailed description of the used methods which may allow repeating analogous experiments or next steps to obtain more quantitative results or address future studies. And also, the honesty to provide the details of the obtained results for other researchers to avoid using the same approach which can lead to same undesirable results.

The conclusions section is very well written and didactic. Overall this is a well written, short, focused and useful paper that reserves publication in Microorganisms.

Figure 1. Use bigger size fonts, especially in the inside box.
Figure 7. Check all labels and indicate units in the axes.
Supplementary Figure 1. Use bigger size fonts for all text.

Author Response

Response to Reviewer

1)The upper image has been increased in size to increase the readability of the font and the middle image has been redone to change the font size to better match the figure legend.

2)An omission of the BurA abbreviation in the legend has been corrected and a description of the axes has been added to the lower left corner of the image to help readers identify the axes without having to reduce the size of the image.

3)All fonts have been bolded and increased in size to help with readability.

Round 2

Reviewer 1 Report

Microorganisms (Manuscript ID: microorganisms-1585421), Comments to the Authors:

Title: Dinoflagellate Phosphopantetheinyl Transferase (PPTase) and Thiolation Domain Interactions Characterized Using a Modified Indigoidine Synthesizing Reporter

Comments

After reading the authors’ response to my comments, I think the revised manuscript can be accepted for publication.